# Impact Energy Release Characteristics of PTFE/Al/CuO Reactive Materials Measured by a New Energy Release Testing Device

**DOI:** 10.3390/polym11010149

**Published:** 2019-01-16

**Authors:** Liangliang Ding, Jingyuan Zhou, Wenhui Tang, Xianwen Ran, Yuxuan Hu

**Affiliations:** College of Liberal Arts and Sciences, National University of Defense Technology, Changsha 410073, China; dingliangliang14@nudt.edu.cn (L.D.); zhoujingyuan12@163.com (J.Z.); ranxianwen@nudt.edu.cn (X.R.); huyuxuan15@nudt.edu.cn (Y.H.)

**Keywords:** PTFE/Al/CuO, reactive materials, impact-initiated energetic materials, sintering process, energy release test, drop hammer

## Abstract

Metal/polymer reactive materials have been studied and applied in a wide range of ways in recent years. This type of material is insensitive under normal conditions but reacts violently and releases a large amount of chemical energy under high-speed impact or high strain rate loading conditions. Compared with conventional explosives, it has better mechanical properties, and its unit mass energy is several times that of TNT. In this paper, PTFE/Al/CuO reactive materials are the main research objects, and we assess the impact energy release abilities of this type of reactive material through experimental research. To this end, eight sets of material formulations are designed, and the effects of particle size, the ratio of PTFE/Al and Al/CuO materials, and sintering on the energy release ability of the reactive materials are investigated. All experiments are carried out based on a self-designed new energy release testing device. The experimental device can measure the pressure time history curve generated by the reactive materials, and the rationality of the pressure time history curve can also be verified by the displacement time curve of the piston. The results show that with an increase in the Al/CuO thermite content, the energy release rate of the reactive material clearly increases, which is attributed to the reaction threshold of Al/CuO being low and because the heat generated can promote the reaction of PTFE/Al. The energy release rate of the nano-scale reactive materials is higher than that of the micron-scale reactive materials because the reduction in particle size results in a larger specific surface area. Thus, the energy required for ignition is lower. The energy release rate of sintered reactive materials is higher than that of unsintered reactive materials, which can be explained by the interfacial area between Al particles and PTFE particles in sintered reactive materials being larger, which makes the reaction more sufficient. The self-designed energy release testing device for the reactive materials and the conclusions obtained in this paper have clear significance for guiding engineering applications.

## 1. Introduction

Reactive Materials (RMs), or Impact-Initiated Energetic Materials, are a class of special materials that were introduced in the 1970s and have been widely applied after more than 30 years of research [1]. There are many types of impact reactive material; the typical examples are intermetallic compounds, thermite, and metal/polymer mixtures. They differ greatly in reaction mechanisms, energy release, and preparation methods. A typical mixture of metal/polymer materials represented by active metal particle-reinforced fluoropolymer materials has received special attention in recent years, resulting in application in weapons [2,3]. These materials are usually composed of two or more non-reactive materials, such as highly active metals and fluoropolymers, which are usually insensitive but react rapidly and violently under high-speed impact or high strain-rate loading conditions, releasing large amounts of chemical energy. In addition, these materials have better mechanical properties than conventional explosives and are sufficiently insensitive to externally-triggered stimuli, such as electricity, heat, and lasers. Their unit mass energy is several times that of TNT. Taking PTFE/Al (73.5/26.5 wt %) materials as an example, the unit mass energy is 3.5 times that of TNT, the energy per unit volume is 5 times that of TNT [4], and the chemical potential released by the reaction is tens of times higher than the kinetic energy of TNT [5]. Therefore, when these materials are used in military applications, such as explosively formed projectiles (EFP) and fragmentation warheads, they can not only damage the target through kinetic energy, but also produce rapid and violent combustion or detonation after the interaction with the target, producing a high amount of pressure and releasing a large amount of heat, thus causing both physical and chemical damage to the target [6]. To this end, this paper mainly uses metal/fluoropolymer reactive materials as research objects to carry out the experimental research.

Research on the enhancement effect of fluoropolymer-based reactive materials on the explosive damage effect began in the 1980s [7]. In recent decades, many scholars have conducted extensive research on such materials. At present, research on fluoropolymer-based reactive materials mainly focuses on the formulation and preparation process, mechanical properties and constitutive relations, critical conditions of impact reactions, and energy release characteristics. In 1997, Wu et al. [8] found that the use of reactive ion assisted interface bonding and mixing (RIAIBM) and annealing surface treatment can promote the mixing of Al and PTFE materials and increase their bond strengths. In 2003, Joshi [9] patented a preparation process of PTFE/Al material with increased strength that allowed it to withstand a sufficiently large launch overload without breaking. Yang et al. [4] studied the preparation process and properties of PTFE/Al materials in 2008. The physical and chemical properties, thermal decomposition properties, and mechanical properties of the materials prepared by this process were studied. In 2016, by studying the influence of the molding pressure of the PTFE material on the post-sintering outcomes, Gamboni et al. [10] discovered that the compressive gas voids generated during the pressing process caused the internal stress to be greater than the crystal cohesive force and formed defects. From 2006 to 2012, Cai and Nesterenko conducted a series of studies on the mechanical properties of Al/PTFE materials and Al/W/PTFE materials, including mechanical and microstructural properties, high strain, strain rate behavior, and failure properties [11,12,13,14], and the effects of the metal particle size on the mechanical properties, strength, failure, and impact of the materials [15,16,17,18].

Ren et al. [19] conducted a comparative study on the mechanical properties of four different proportions of PTFE/Al/W materials in 2016. The author divided the quasi-static compression stress-strain curves into three stages: elasticity, hardening, and destruction. In 2007, Rosencrantz [20] described, in detail, the characterization and modeling methods of PTFE reactive materials in his Master’s thesis and determined the JWL equation of state and the parameters of the Al/PTFE reactive materials. He created an equation of state (EOS) and applied it to LS-DYNA. Rafenberg [21] proposed a JCP constitutive model for PTFE/Al (26.5/73.5 wt %) materials in 2008 which correlates the yield strength, plastic strain, strain rate, and temperature values. Jiang et al. [22] studied the strength model and EOS of PTFE/Al reactive materials in 2012. Based on the theory of Meyers and the Voigt–Reuss–Hill mixing law, the relevant parameters of the JWL equation of state of the material were determined. In 2005, Lee et al. [23] utilized the direct impact, indirect impact, and two-step impact methods to study the impact-induced reaction of PTFE/Al reactive materials. The results indicated that the materials reacted at the initial impact stage, but the main reaction occurred after the broken and cracked sample collided with the second impact surface. Zamkov et al. [24] carried out research on the reaction rates and ignition reactions of reactive materials for nanoparticles. The research showed that nano-scale aluminum significantly increases the reaction rate of Al/PTFE materials.

Denisaev et al. [25] studied the initiation reaction of Al/PTFE film using a drop weight test in 2008. The results indicated that Al/PTFE material is a brittle, deformed material and will break down rapidly after a certain level of pressure elastic compression, and some local initial ignition reactions occur on the shear plane. Many scholars at the Naval Surface Warfare Center have studied the impact ignition reactions of reactive materials using a light gas gun and a Hopkinson bar. Ames et al. [26] discovered the detonation-like energy release of Al/PTFE formulations in 2002. Subsequently, in 2003, Ames published a test method for closed containers which can be used to study the energy release of the impact reactions of Al/PTFE formulations. In 2004, McGregor et al. [27] discovered that high porosity (40% compactness) PTFE/Al samples ignited after the first shock wave through the light gas gun impact test, but when the secondary shock wave pressure was high, the material underwent a re-ignition reaction. In 2005, Ames [28] conducted a theoretical analysis of the designed test method, showing the relationship between the pressure inside the container and the amount of energy released, and proposed the concept of the energy release rate. In 2006, Ames [29] measured the energy release efficiencies of Al/PTFE, Zr–THV (mixture of three different fluoropolymers: TFE, HFP (hexafluoropropylene), and VF (vinylidene fluoride)), Ta–THV and Hf–THV at 1200 m/s, 1800 m/s, and 2400 m/s. In 2007, Ames [30] introduced and published a standard test method in open literature.

According to the above-mentioned literature, we know that a lot of work has been done on the preparation sintering process, mechanical properties, and energy release ability tests of reactive materials. These works have achieved remarkable results which lay the foundation for this paper. Based on the background of engineering research, this paper hopes to obtain a reactive material formulation with a higher energy release ability. Since the densities of Al and PTFE are relatively low, the density of the reactive material made of these two materials is also low. In order to ensure the density of the reactive material while also taking into account the energy release ability of the reactive material, CuO powder is added to the Al and PTFE powders. Due to the density of CuO being relatively high, the combustion reaction rate of the thermite Al/CuO is higher than that of the Al/PTFE. Therefore, the addition of CuO powder not only adjusts the density of the reactive material but also promotes the chemical reaction between the components of the reactive materials. Thus, the main raw materials of the PTFE-based reactive material designed in this paper include Al, PTFE, and CuO powders. In order to obtain the reactive material formulation with a higher energy release ability, several formulations were designed based on the particle size distribution and reaction efficiency. Drawing on the energy release testing device designed by Ames [29] and the drop hammer test system, a new type of energy release testing device was self-designed to determine the optimal formula. It is hoped that the self-designed energy release testing device and the obtained conclusions can provide a reference for engineering testing and structure design.

## 2. Preparation of the Reactive Materials

The main raw materials required for the designed reactive materials include PTFE, Al, and CuO powders, as shown in Figure 1. The addition of CuO powder is mainly used to promote the chemical reaction between the reactive materials and to increase the density of the reactive materials. The particle sizes of the raw materials include PTFE powder (350 nm and 35 μm), Al powder (50 nm and 5 μm) and CuO powder (50 nm and 10 μm). The instruments required for the experiment include a tablet press, an ultrasonic disperser, a tubular vacuum sintering furnace, a mixing mixer, a vacuum drying oven, and an electronic scale. The preparation and sintering process of the PTFE-based reactive materials is shown in Figure 2.

### 2.1. Formulation of the Reactive Materials

The reaction equation for PTFE and Al in an inert gas is as follows:(1)$${4\mathrm{Al}(s)+3-(C}_{2}F_{2}{)-(s)=4\mathrm{AlF}}_{3}\left(s)+6C(s\right)$$

The reaction equation for Al and CuO is as follows:(2)$${2\mathrm{Al}+3\mathrm{CuO}=\mathrm{Al}}_{2}O_{3}+3\mathrm{Cu}$$

According to the above reaction equations, the reaction mass ratio of Al and PTFE is 26.5:73.5, and the reaction mass ratio of Al and CuO is 18.5:81.5. Considering the particle size distribution and the reaction rate, this paper attempts to design the material formulation with PTFE/Al and Al/CuO as independent units. First, PTFE/Al and Al/CuO were respectively arranged in their reaction mass ratios. Then, PTFE/Al and Al/CuO were mixed together in different proportions to form different material formulations. Finally, a series of material formulations was designed, as shown in Table 1.

The material compositions and ratios of the formulations #1 and #2, #3 and #4, #5 and #6, and #7 and #8 are the same in pairs, but the particle sizes are different. The reactive material is based on the PTFE/Al combination. Since the energy release condition of PTFE/Al is high, the energy release efficiency is low, and copper thermite was added to promote the energy release. In addition, in order to effectively improve the energy release efficiency, the addition of the nanoparticle copper thermite was considered. Nanomaterials can reduce the distance between molecules, depress the activation energy of the material reaction, and accelerate the reaction rate. Al/CuO thermite has a relatively high reaction burning rate and it can achieve the highest combustion reaction rate after nanocrystallization. Moreover, it has high sensitivity and is easy to use to stimulate the reaction, and so, in theory, it is a feasible scheme to use to improve the energy release ability of materials.

### 2.2. Pretreatment of the Raw Material Powder

PTFE is an extremely hydrophobic organic high molecular polymer, and it has poor affinity with metal materials such as Al, which adversely affects the overall mechanical properties of the composite material. Therefore, it is necessary to optimize the surface of the PTFE. The interface modifiers and mechanical reinforcing agents commonly used in polymer matrix composites mainly include coupling agents, bonding agents, crosslinking agents, and curing agents.

The coupling agent is a kind of plastic additive, also known as a surface modifier, which can improve the interface performance between the synthetic resin and the inorganic filler or be used as a reinforcing material. The primary characteristic of the molecular structure in the coupling agent is that the molecule contains two groups with different chemical properties. One is a hydrophilic group, which easily reacts with the surface of the inorganic substance, and the other is an organophilic group, which can chemically react with the synthetic resin or form hydrogen bonds and dissolve therein. According to the characteristics of the raw materials, the silicon coupling agent is selected to pretreat the surface of the metal particles, and the amount of silicon coupling agent used is generally 0.5~2.0% of the amount of filler.

The specific operation steps are as follows: First, a certain amount of coupling agent is dissolved in the absolute ethanol, and then the metal powder is put into the organic solvent and allowed to stand for 1 h; after that, the organic solvent is heated and stirred until it evaporates completely, and then it is put into the vacuum drying box and dried for more than 6 h. At this point, the pretreatment of the metal material has been completed, followed by mixing with the organic material. A certain amount of PTFE powder is mixed with the surface pretreated metal powder in the organic solvent and stirred for about 10 h. Then, the organic solvent is heated until it evaporates completely. Finally, the powder material is put into the vacuum drying oven again and dried for at least 12 h until it is thoroughly dried.

### 2.3. Forming of the Reactive Materials

Unlike ordinary plastic materials, PTFE has a relatively high viscosity after it has been heated and melted, and thus it does not have sufficient fluidity and will basically maintain its original shape. In addition, PTFE is sensitive to shear force and is easy to crack after melting at a high temperature, so it is not suitable for the conventional thermoplastic forming process. To this end, it is necessary to select a more versatile compression molding method and to ensure the formed sample is as dense as possible during processing so that the sample will not crack or delaminate due to the internal residual stress of the sintering process. Therefore, after repeated attempts, the cold pressing process is used to compress the samples.

After determining the forming process, according to the requirements and characteristics of the experiments, a set of simple and intuitive pressing molds was designed, and their structure diagram and physical map are shown in Figure 3. The whole pressing mold consists of three parts: the punch, the sleeve, and the base. To facilitate the filling of materials into the mold, a bell mouth is milled on the upper end of the sleeve. At the same time, in order to reduce the cracks caused by the expansion during the demolding process, a small chamfer is also milled at the lower end of the sleeve. It is worth noting that due to the poor fluidity of PTFE, the material will remain on the inner wall of the sleeve during the compression process, and the inner wall of the sleeve needs to be cleaned over time. If the cleaning is not timely, the friction between the punch and the inner wall is greatly increased, making the punch and the base difficult to remove. The effect is more obvious when the pressing pressure is higher. In addition, the coaxiality of the mold should be retained as much as possible during the pressing process.

The pressing process of the sample is as follows. First, the mixed powder material is put into the mold as far as possible to reduce the delamination caused by the pre-extrusion. Then, the pressure and temperature should be controlled in a stable range during the pressing process. The forming pressure is 60~100 MPa, and the temperature is room temperature. Finally, after the mold is pressed into the position, the pressure is released after half a minute of holding pressure, and the sample is taken out and allowed to stand for 24 h to reduce the prestress inside the material.

### 2.4. Sintering of the Reactive Materials

The sintering process is a key step in changing the strength of PTFE. The most affected material during the sintering process is the PTFE, which undergoes a series of physical and chemical changes. When the temperature rises above the melting point of PTFE, 327 °C, the molecular crystallization of the polymer gradually transforms into the amorphous, dispersed, individual resin particles, which are melted into a continuous whole using interdiffusion. After cooling and crystallization, the polymer molecules gradually change from amorphous to crystalline. PTFE plays a major role in the mechanical properties of PTFE-based reactive materials, so the sintering temperature is mainly based on PTFE. The melting temperature of PTFE is near 327 °C, and the material begins to vaporize gradually after exceeding 400 °C. When the sintering temperature rises above 380 °C, the density of sintered materials decreases rapidly due to vaporization and decomposition, and so the sintering temperature in this paper was selected to be between 360 and 380 °C. In order to prevent the material from reacting during the sintering process, it should be sintered in a vacuum or an inert gas atmosphere. The heating rate of sintering is 60 °C/h, and the temperature is maintained at around 380 °C to start the heat preservation. The holding time is determined by the sample mass, usually at 1~6 h. Thereafter, the temperature is lowered to 275 °C at a rate of 52 °C/h, and the heat should be kept for 3 h. Finally, the sintering device is turned off and allowed to naturally cool down to the room temperature. The sintering temperature curve in this paper refers to the sintering temperature curves of Joshi [9] and Nielson [31], and it is adjusted according to the actual situation. The sintering temperature curve is shown in Figure 4.

In this paper, an open vacuum/atmosphere tubular sintering furnace was used for sintering. It mainly consists of the upper furnace body, lower furnace body, control box, quartz furnace tube, and vacuum sealing accessories (special for vacuum/atmosphere tube furnace), as shown in Figure 5. The tubular electric furnace uses 0Cr27Al7Mo2 electrothermal alloy as the heating element. The furnace adopts a PMF (Polycrystalline Mullite Fiber) ceramic fiberboard with low thermal conductivity and less heat storage, and the working temperature in the furnace can reach 1200 °C. In addition, it has the characteristics of fast heating and a uniform furnace temperature. It also functions with programmed temperature and vacuum pump controls, and the temperature control system uses the programmed control instrument + K type thermocouple. Through the feedback mechanism, the temperature in the furnace can be measured, displayed, and controlled so that the temperature in the furnace can run automatically according to the pre-set heating curve with high control accuracy. The protection system adopts both over-temperature and leakage protections, and the double protection provides a reliable guarantee for the safety of users. The over-temperature protection can ensure that the power supply of the heating element can be automatically cut off when the actuator fails and the furnace temperature cannot be controlled. The leakage protection ensures that the total power of the equipment can be automatically cut off when the insulation of the electric furnace is damaged. Before pumping the vacuum tube, it is necessary to check whether the suction port and the vacuum control system are properly connected. When vacuuming, the vacuum pump should be turned on first, and then the heating is turned on when the inside of furnace tube is under the vacuum.

According to the requirements of different test types, several sets of samples of different sizes were designed, including ø10 × 3 mm, ø10 × 10 mm, and ø10 × 30 mm. The pre-sintering and post-sintering states of the specimens of different sizes are shown in Figure 6.

## 3. Design of the Energy Release Testing Device

Existing conventional energetic materials, such as explosives, have a variety of relatively complete testing methods for mechanical properties and energy release ability. As new types of energetic material are designed, the series of testing methods for reactive materials becomes imperfect. Therefore, this paper hopes to draw on the testing methods of conventional energetic materials and the existing testing methods of reactive materials to form an innovative design in accordance with the situation.

### 3.1. Design Ideas for the Energy Release Testing Device

The energy release of reactive materials requires shock initiation and a high strain rate of plastic deformation or fracture by external forces. Therefore, the release testing method of the reactive material cannot completely refer to the method used for conventional energetic materials. For this reason, Ames [29] designed a dynamic energy release testing method which can quantitatively characterize the energy release ability of the reactive materials. The schematic diagram of the test principle is shown in Figure 7.

The energy release testing device designed by Ames adopts a sealed cylindrical chamber with a thin target plate at one end. A hardened steel anvil is designed inside the chamber which provides an impact surfaces for the projectiles. The reactive material projectiles often lose part of their mass when they penetrate the thin surface target plate, and, at the same time, some reactions begin to occur to some extent. Due to the low strengths of the reactive materials, the remaining materials generally impact the hardened steel anvil in the form of a loose powder which will produce an impact-initiated reaction.

The initial reaction process is similar to detonation because the reaction of the reactive materials is relatively fast and the shock wave propagates through the whole chamber rapidly, but only a small amount of the reactive material participates in the detonation reaction. As time progresses, the remaining reactive materials continue to react. The initial reaction time is generally in the range of 1~10 μs, and the later-time (or “afterburn”) reaction time is generally in the range of 1~10 ms. As far as these two reaction processes are concerned, the first reaction is similar to the detonation reaction, and the pressure changes rapidly in a very short period of time. The afterburn reaction process is a relatively slow process; the time range of this process is relatively wide and the pressure is relatively stable, and the pressure generated was called “quasi-static” pressure by Ames [29]. The “quasi-static” pressure is basically the average of the explosion pressure fluctuations. The difference between the two pressure phenomena is given in Figure 8.

The reaction process of the reactive material in the chamber can be regarded as an adiabatic reaction process. Combined with the equation of state of the ideal gas, the following relation can be obtained:(3)$$\Delta P=\frac{\gamma -1}{V}\Delta E$$where ∆*P* is the peak quasi-static pressure, *γ* is the specific heat ratio of the gas in the chamber, *V* is the chamber volume, and ∆*E* is the total energy of the reactive material deposited into the chamber. Note that the total energy value here includes the kinetic energy and the energy released by the chemical reaction.

For explosives, the degree of difficulty with respect to detonation under the action of external energy is usually called the sensitivity of explosives. The sensitivity of explosives is generally divided into thermal sensitivity, impact sensitivity, shock wave sensitivity, electrostatic spark sensitivity, and so on. Taking the impact sensitivity as an example, there are various methods for expressing the impact sensitivity, such as the explosion percentage method, the upper and lower limit method, and the characteristic falling height method. Most of these testing methods are based on the drop hammer test system. The structural schematic and physical diagram of the drop hammer test system are shown in Figure 9.

The above two test systems have their own advantages. Ames’ test system can quantitatively test and characterize the energy release ability of reactive materials, usually requiring a higher impact velocity. The drop hammer test system quantitatively compares the energy release ability of reactive materials, and the impact velocity is usually lower. Therefore, we hope to design a new type of energy release testing device that is suitable for the test requirements of this paper by drawing on the advantages of the above two test systems. The specific requirement is to quantitatively measure and characterize the energy release ability of reactive materials at a lower impact velocity.

### 3.2. Engineering Design and Installation of the Energy Release Testing Device

According to the test requirements, we designed an energy release testing device based on the drop hammer test system which can measure the energy release effect of reactive materials under the impact of the drop hammer. The engineering entity diagram and engineering perspective diagram of the energy release testing device are shown in Figure 10. The internal dimensions of the chamber are 100 mm × 100 mm × 110 mm, the size of the chopping block is 50 mm × 50 mm × 45 mm, the inner diameter of the guide sleeve is 30 mm × 100 mm, and the size of the impact plunger under the upper-end cover is 30 mm × 52 mm + 40 mm × 8 mm.

Since the clearance reserved for each channel is small, the testing device can be approximated as a quasi-closed container. The working principle of this energy release testing device is that when the drop hammer hits the impact plunger, the impact plunger will further hit the reactive material sample placed on the chopping block, thus stimulating the reaction of the reactive materials. The high-pressure gas and products produced by the reaction will be released via the piston pipe, thus promoting piston movement. At this time, the pressure sensor on the back wall of the chamber can measure the pressure change inside the chamber. In addition, we can also deduce the displacement of the piston according to the movement of the piston.

When studying the energy release characteristics of reactive materials, we can not only analyze the energy release effect of reactive materials by measuring the change in pressure over time with the pressure sensor, but also, we can deduce the functional force of the reactive materials when they react inside the chamber by analyzing the piston movement with high-speed photography. These two results can be verified and complement each other so that the energy release ability of the reactive materials can be tested more accurately. The layout diagram of the energy release test is shown in Figure 11.

As can be clearly seen from Figure 11, a wooden board with coordinate grid paper is pasted on the side of the energy release testing device, and the coordinate grid paper is used to calibrate the displacement of the piston; the high-speed photography and light source are placed directly opposite the coordinate grid paper, that is, the high-speed photography is placed perpendicular to the coordinate grid to record the trajectory of the piston after the reactive material reacts. The specific operation process is as follows: (1) The lower base plate of the testing device is fixed on the base of the drop hammer device; (2) the sample is placed in the center of the chopping block, and then the upper cover is closed and tightened with screw bolts; (3) the impact plunger and piston are inserted into the designated position; (4) the drop hammer test system and high-speed photography are synchronously triggered and record the data; (5) the drop hammer test system and high speed photography equipment are reset; and (6) the energy release testing device is opened and the chamber and guide sleeve are cleaned.

## 4. Analysis and Discussion of the Experiment Results

The chemical reaction heat of the two groups of materials involved in this paper can be obtained using chemical theoretical analysis. The reaction heat of Al/CuO (18.5:81.5) is 4077.438 J/g, and the reaction heat of PTFE/Al (73.5:26.5) can be calculated according to Gas’s law. The reaction heat is proportional to the amount of the substance, and it is related to the initial state (reactants) and final state (products), but not to the pathway of the reaction. That is to say, if a reaction can be carried out step by step, the sum of the reaction heat of each step reaction is the same as that of the reaction heat when the reaction is completed in one step. The standard molar enthalpy of the formation of the simple substance is zero, and those of PTFE and AlF_3_ are −854 kJ/mol and −1510.4 kJ/mol, respectively. Thereby, the reaction heat can be calculated according to the chemical reaction equation, and the chemical reaction equation of PTFE/Al is as follows:(4)$${4\mathrm{Al}(s)+3-(C}_{2}F_{2}{)-(s)=4\mathrm{AlF}}_{3}\left(s)+6C(s\right)$$

Thus, the enthalpy change of reaction can be obtained as follows:(5)$$\Delta_{r}H=4\Delta H(AlF_{3})-3\Delta H(PTFE)=4\times (-1510.4)-3\times (854)=-3479.6\;\mathrm{kJ}$$

Thereby, the reaction heat value can also be obtained:(6)$$Q_{r}=-\Delta_{r}H/M=\Delta_{r}H/\left(3M_{\mathrm{PTFE}}+4M_{\mathrm{Al}}\right)=3479.6/\left(3\times 100+4\times 27\right)=8.53\;\mathrm{kJ}/g$$where *M*_PTFE_ and *M*_Al_ are the relative molecular masses of PTFE and Al, respectively.

Based on the proportional relationship of the chemical reactions of the two groups of materials, the theoretical energy per unit mass of the corresponding reactive materials can be obtained. Then, by testing and analyzing the energy actually released after the reaction of the reactive materials in the experiment, the energy release rate of the reactive materials can be obtained. The experiments in this paper use the drop hammer to strike the impact plunger to stimulate the reactive materials to react, and then the pressure in the container rises and the piston in the pipe is pushed outward. The piston is made of Al with a weight of 69.86 g and a diameter of 30 mm. The mass of the drop hammer is 10.0 kg and the maximum stroke of the drop hammer system is 2.5 m, so the whole impact process has a low-speed impact. In view of the characteristics of this device, the size of the reactive material samples used in this paper is ø10 × 3 mm.

### 4.1. Impact Energy Release Test of Micron-Scale PTFE-Based Reactive Materials

According to the working conditions shown in Table 1, the formulations of #1, #3, #5, and #7 are all micron-scale PTFE-based reactive materials. In this section, the energy release characteristics of sintered and unsintered PTFE (μm)/Al/CuO reactive materials are discussed, and the energy release results measured by the pressure sensors are compared with those derived from piston motion. In addition, the height of the drop hammer is set to 2.0 m in this section.

The pressure time history curve measured by the pressure sensor of the unsintered PTFE (μm)/Al/CuO reactive materials (#1, #3, #5, #7) under the impact of the drop hammer with a height of 2.0 m is shown in Figure 12.

During the experiments, it was found that the #1 reactive material did not react substantially, while the other three kinds of reactive material (#3, #5, #7) caused the pressure in the container to rise slightly, as shown in Figure 12. This phenomenon indicates that only a very small portion of the reactive materials reacted for the unsintered PTFE-based reactive materials. That is to say, under this drop weight condition, the energy release rate of the micron-scale PTFE-based reactive material was very low. Although the overpressure value was small, we can still see that the three sets of overpressure peaks corresponded to the following relationship: #7 > #5 > #3. This shows that the increase in thermite content helps to promote the reaction of the reactive materials. More importantly, the overpressure in the container is not enough to push the piston to overcome the frictional force of the system, so the displacement time curve cannot be reversed. In addition, raising the drop hammer to the maximum height of 2.5 m did not significantly increase the energy release of the reactive materials.

However, the sintered micron-scale PTFE-based reactive material is capable of reacting and driving the piston, and the initial conditions of the drop hammer are consistent with the previous ones. It can be seen from Figure 8 that after the reactive material reacted in the container, the shock wave was first generated, and then the quasi-static pressure formed in the container. In engineering research, the main concern is the quasi-static pressure value of the second half. Therefore, when dealing with the measured pressure time history curve, the data of the initial shock wave was not analyzed, and only the second half of the data was retained for analysis. By using the data analysis software Origin (OriginLab, Northampton, MA, USA) to process the measurement curves, the pressure time history curves of the sintered micron-scale PTFE-based reactive materials measured by the pressure sensor were obtained, as shown in Figure 13.

As can be seen from Figure 12 and Figure 13, the energy release effect of the sintered micron-scale PTFE-based reactive materials significantly improved compared with the unsintered micron-scale PTFE-based reactive materials. This is because the increase in temperature in the sintering process makes the contact between PTFE and Al powder in sintered reactive material more sufficient. Therefore, it reacts more easily under the external impact event. In addition, by comparing Figure 8 and Figure 13, it can be seen that the quasi-static pressure value in Figure 8 finally tended to reach a stable value, and the pressure value in Figure 13 gradually decreased after reaching the peak value. This is because a movable piston was designed in the energy release testing device. After the piston slides out of the guide sleeve, the pressure in the container will inevitably drop rapidly and finally, it will become consistent with the atmospheric pressure. Therefore, the peak value of the overpressure can be approximated as a quasi-static pressure value for the analysis.

The reaction of reactive materials in the chamber will cause pressure changes, and the reaction degree and energy release rate of the reactive materials can be reflected by the pressure value. Therefore, in order to be able to quantitatively compare the energy release rates of several groups of formulations, the energy release rate of the reactive materials is defined herein:(7)$$\eta =\frac{\bar{\Delta P}}{\Delta P^{*}}$$where *η* is the energy release rate of the reactive materials, $\bar{\Delta P}$ is the experimentally-measured average value of the overpressure, and $\Delta P^{*}$ is the theoretical pressure change, which can be obtained according to Formula 3. Thus, the energy release rate of the four sets of sintered formulations (#1, #3, #5, #7) can be obtained as shown in Table 2.

It can be seen from Figure 13 and Table 2 that as the content of copper thermite increases, the energy released by the reactive material increases. That is to say, the order of the energy release rate is *η*_#7_ > *η*_#5_ > *η*_#3_ > *η*_#1_. Generally speaking, the reaction efficiency of the sintered micron-scale reactive materials is relatively low, and the maximum efficiency is less than 21%. Although the exotherm release per unit mass of Al/CuO thermite is lower than that of the PTFE/Al material, the theoretical total energy will decrease after adding Al/CuO thermite, but the excitation sensitivity of Al/CuO thermite is higher than that of the PTFE/Al reactive material, and the reaction propagation speed is faster. Therefore, the addition of Al/CuO thermite can improve the reaction efficiency of the reactive materials, making it release more energy instead.

At the same time, the piston displacement time curves corresponding to different formulations can be obtained using high-speed photography, as shown in Figure 14. Taking formulation #7 as an example, the piston movement state captured with high-speed photography is shown in Figure 15.

After the reactive materials react under the external impact, the piston moves outward under the action of the chamber pressure. Since the piston mass is always constant, the length of time required for sliding and flying displacement can reflect the pressure inside the chamber, which can reflect the amount of energy released by the reactive materials. The length of the entire guide sleeve is 100 mm, so it can be seen from Figure 14 that the time required for the piston to slide out of the guide sleeve is *t*_#7_ < *t*_#5_ < *t*_#3_ < *t*_#1_. The order of magnitude of the energy release rate obtained above can also be verified, namely *η*_#7_ > *η*_#5_ > *η*_#3_ > *η*_#1_. The mutual verification between the two test methods indicates that the energy release testing device designed in this paper is feasible. In addition, from the release process of formulation #7 shown in Figure 15, we can see a large amount of black smoke following the piston and a small amount of flare. This indicates that the reactive materials reacted inside the cavity, but the reaction was not sufficient. After analysis, the black smoke mainly contained unreacted reactive material powder and some formed products.

### 4.2. Impact Energy Release Test of Nano-Scale PTFE-Based Reactive Materials

According to the formula design in Table 1, the formulations of #2, #4, #6 and #8 all use nano-scale powder materials, which correspond to #1, #3, #5 and #7 in the formulation ratio. Similarly, the time history curve of the impact energy release pressure of the unsintered nano-scale PTFE-based reactive materials can be obtained, as shown in Figure 16.

As is apparent from the comparison of Figure 12 and Figure 16, both are unsintered reactive materials. Since the particle sizes of the matrix powder constituting the nano-scale reactive materials are smaller than those of micron-scale reactive materials, the nano-scale reactive materials are more likely to excite the reaction, and the energy release rate is also greatly improved. This is mainly because the nano-powders have a larger specific surface area and require less energy to stimulate the reaction. At the same time, in the process of drop hammer impact, energy is transferred from the drop hammer to the sample, and the hot spots are more easily generated in the nano-scale samples. When the material reaction at the hot spots releases enough energy, it will cause the reaction at the hot spots to continue and will cause the reaction of the entire material. In addition, by comparing Figure 15 and Figure 16, the energy release capacity of the unsintered nano-scale PTFE-based reactive material is similar to that of the sintered micro-scale PTFE-based reactive material. Similarly, the energy release rate of the four sets of unsintered formulations (#2, #4, #6, #8) can be obtained as shown in Table 3.

It can be seen from Figure 16 and Table 3 that as the content of copper thermite increases, the energy released by the reaction material increases, and the order of the energy release rate is as follows: *η*_#7_ > *η*_#5_ > *η*_#3_ > *η*_#1_. At the same time, the piston displacement time curve corresponding to different formulations can be obtained using high-speed photography analysis, as shown in Figure 17. It can also be seen from Figure 17 that the order of magnitude of the energy release rate is as follows: *η*_#7_ > *η*_#5_ > *η*_#3_ > *η*_#1_.

Similarly, the nano-scale PTFE-based reactive materials were also sintered. Their energy releasing ability was tested, and the energy release rates are shown in Table 4. Taking formulations #2 and #8 as examples, the sintered samples and their corresponding typical pressure time history curves are shown in Figure 18 and Figure 19, respectively.

It can be seen from Table 4 that the energy release of the sintered nano-scale PTFE-based reactive materials is considerable, and the energy release rate is much larger than that of the unsintered nano-scale PTFE-based reactive materials and the sintered micron-scale PTFE-based reactive materials. In addition, it can also be seen that there is a distinct negative pressure zone in Figure 19 which is due to the excessive expansion of the internal pressure in the chamber when the piston moves outward, and then the internal pressure in the chamber gradually becomes consistent with the external atmospheric pressure area under the action of sparse air waves. Therefore, it can be approximately stated that the reaction of the sintered nano-scale PTFE-based reactive material is similar to the detonation of the conventional explosives.

In order to more intuitively compare the energy release ability of the sintered nano-scale PTFE-based reactive materials, the motion states of the piston captured using high-speed photography corresponding to formulations #2 and #8 are shown in Figure 20.

When comparing Figure 20 with Figure 15, it can be found that the reaction produced less black smoke and a larger flare in Figure 20, which indicates that the reaction of the nano-scale PTFE-based reactive materials was more sufficient than that of the micron-scale PTFE reactive material. In addition, by comparing and analyzing formulations #2 and #8 in Figure 20, it can be found that #8 produced a large amount of fire and less black smoke, which verifies the conclusion in Table 4 that formulation #8 has the highest energy release rate.

## 5. Conclusions

In this paper, the PTFE/Al/CuO reactive materials were used as the research object, and the effects of particle size, the ratio of PTFE/Al and Al/CuO materials, and sintering on the energy release ability of the reactive materials were investigated. Based on the principle of the drop hammer test system and Ames’ energy release testing device, a new type of reactive material energy release testing device based on the drop hammer system was self-designed. Eight groups of different material formulations were designed, the energy release tests were carried out respectively, and the pressure time history curve and energy release rate were obtained. By analyzing the obtained test results, the following conclusions can be drawn:(1)With the increase of the Al/CuO thermite content, the energy release rate of the reactive materials increases significantly. Taking unsintered nano-scale reactive materials #2 and #8 as examples, the energy release rates of #2 and #8 were 6.44% and 21.61%, respectively. That is, the energy release rate of #8 was 3.35 times that of #2. However, the reaction heat per unit mass of Al/CuO thermite is lower than that of PTFE/Al, which will result in a theoretical decrease in the total energy per unit mass of the reactive materials after the addition of Al/CuO thermite. However, the reaction threshold of Al/CuO thermite is relatively low compared to that of PTFE/Al and reacts more easily under the low-speed impact of the drop hammer, thereby promoting the reaction of PTFE/Al, and finally, the overall energy release rate of the reactive materials improves.(2)Under the external impact, the energy release ability of the nano-scale reactive materials with the same formulation is significantly better than that of the micro-scale reactive materials. This is attributed to the larger specific surface area of the nano-scale particles than the micron-scale particles, which makes the initial energy needed for the reaction lower and the reaction more sufficient. When the reaction releases enough energy, it will cause the reaction at the hot spots to continue, thus causing the reaction of the entire materials.(3)The energy release rate of the sintered reactive materials under the same impact conditions is higher than that of the unsintered materials. The nano-scale reactive materials especially have a relatively high energy release rate after sintering. Taking sintered nano-scale reactive material #8 as an example, the energy release rate was able to reach 63.63%. The initial melting temperature of PTFE is 327 °C. When the temperature reaches 400 °C or above, it undergoes a depolymerization degradation reaction. Its main chain is broken, and a large amount of small active molecules, such as tetrafluoroethylene, are formed. Since the particle size of the nano-powders is much smaller than that of the micron-powders, the contact between the particles of the sintered nano-scale reactive materials is more sufficient. When the reaction threshold is reached, the Al particles readily react with the active small molecules.(4)Based on the above conclusions, for the PTFE/Al/CuO reactive materials designed in this paper, the energy release rate of reactive materials can be improved by refining the particle size, adjusting the ratio between PTFE/Al and Al/CuO, and adjusting the sintering process to prepare reactive material formulations that meet the engineering needs. The self-designed energy release testing device and the conclusions obtained in this paper can serve as references for future research.

## Figures and Tables

**Figure 1 polymers-11-00149-f001:**
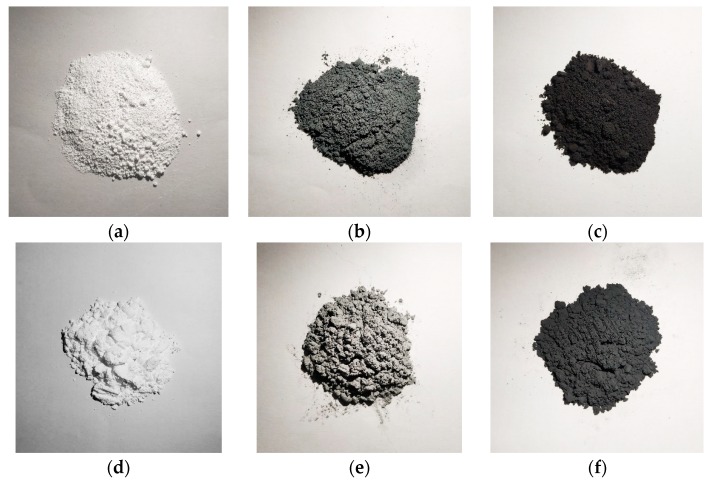
The main raw materials required for the designed reactive materials: (**a**) PTFE (350 nm); (**b**) Al (50 nm); (**c**) CuO (50 nm); (**d**) PTFE (350 μm); (**e**) Al (5 μm); (**f**) CuO (10 μm).

**Figure 2 polymers-11-00149-f002:**
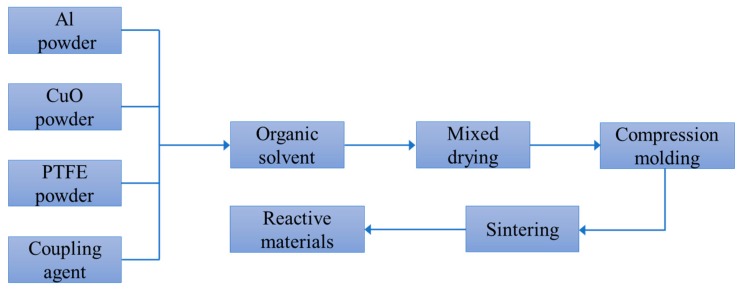
The preparation and sintering process of the PTFE-based reactive materials.

**Figure 3 polymers-11-00149-f003:**
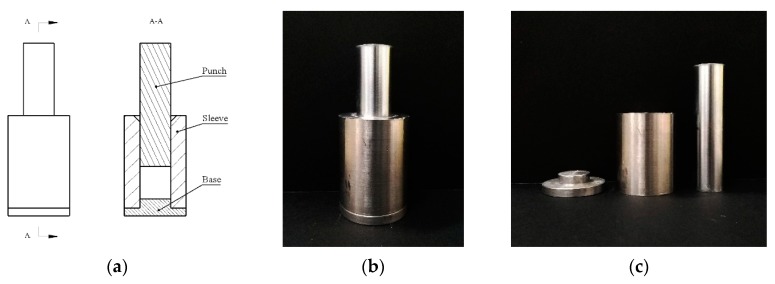
The structure diagram and physical map of the pressing mold: (**a**) the structure diagram; (**b**) the assembly drawing; (**c**) the part drawing.

**Figure 4 polymers-11-00149-f004:**
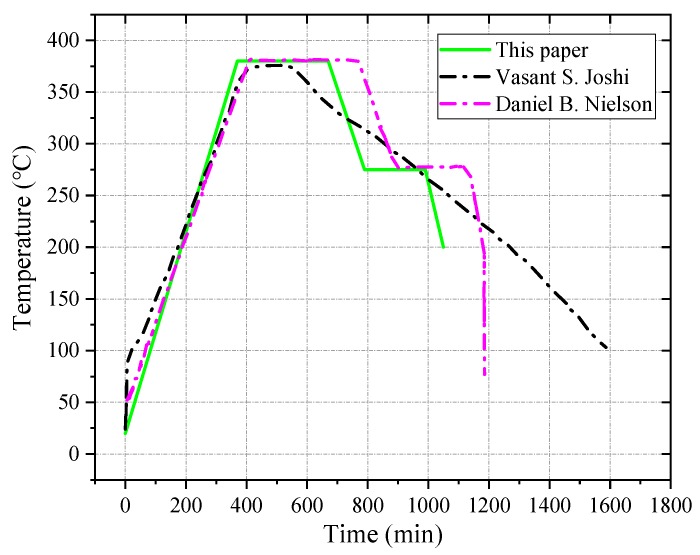
The sintering temperature curve of reactive materials.

**Figure 5 polymers-11-00149-f005:**
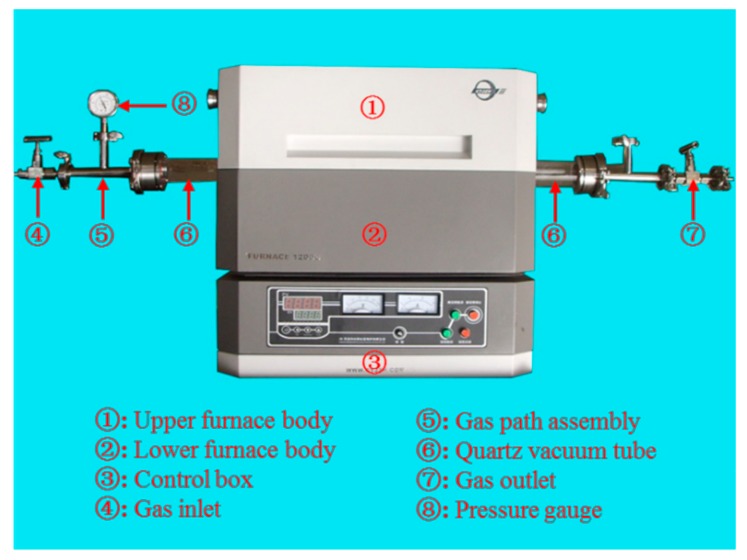
A physical drawing of the tubular vacuum sintering furnace.

**Figure 6 polymers-11-00149-f006:**
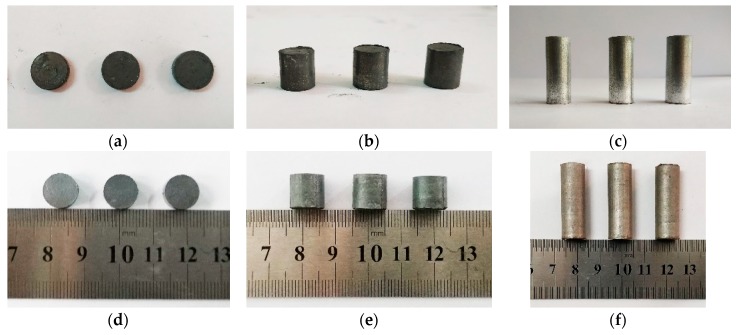
The pre-sintering and post-sintering states of the specimens of different sizes: (**a**) pre-sintering state (ø10 × 3 mm); (**b**) pre-sintering state (ø10 × 10 mm); (**c**) pre-sintering state (ø10 × 30 mm); (**d**) post-sintering state (ø10 × 3 mm); (**e**) post-sintering state (ø10 × 10 mm); (**f**) post-sintering state (ø10 × 30 mm).

**Figure 7 polymers-11-00149-f007:**
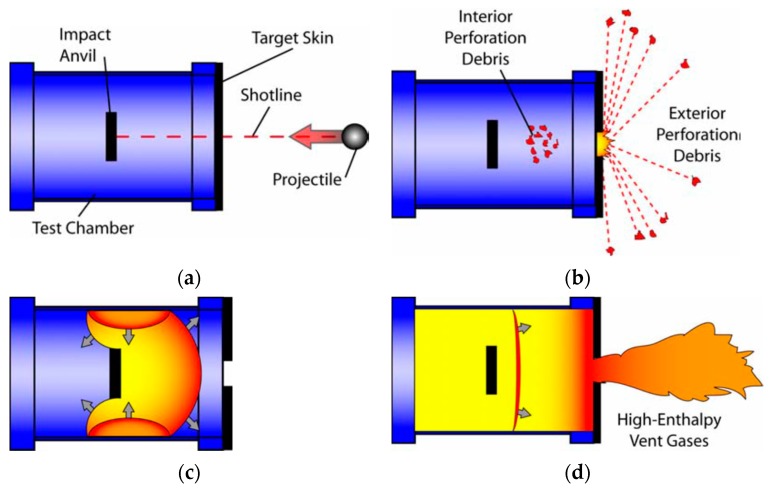
Schematic diagram of the test principle: (**a**) test chamber setup; (**b**) post-perforation flight; (**c**) development of the reflected shocks; (**d**) global burn.

**Figure 8 polymers-11-00149-f008:**
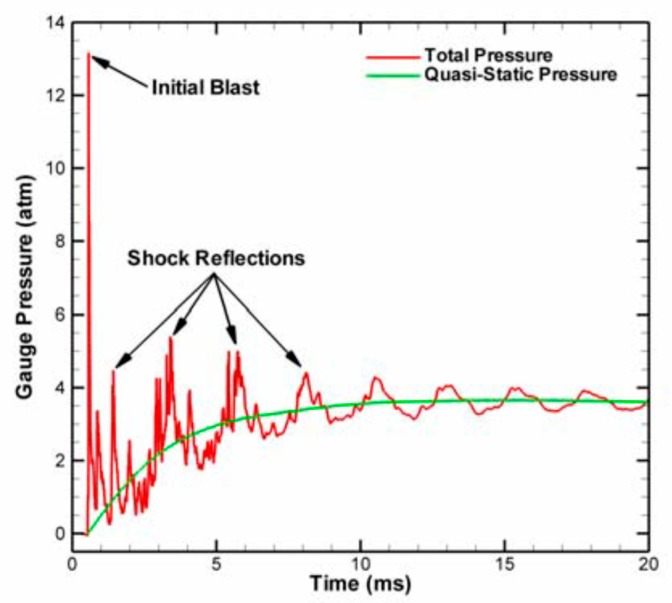
Difference between the blast pressure and the quasi-static pressure.

**Figure 9 polymers-11-00149-f009:**
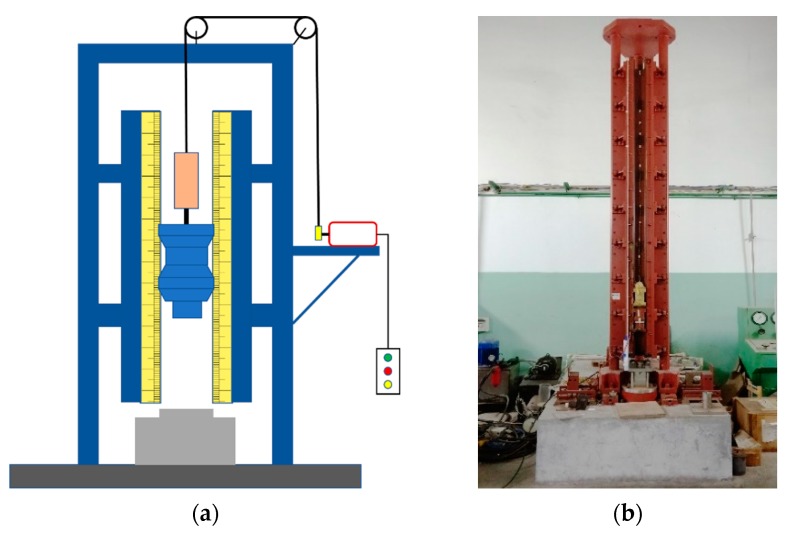
Structural schematic and physical diagram of the drop hammer test system: (**a**) structural schematic; (**b**) physical diagram.

**Figure 10 polymers-11-00149-f010:**
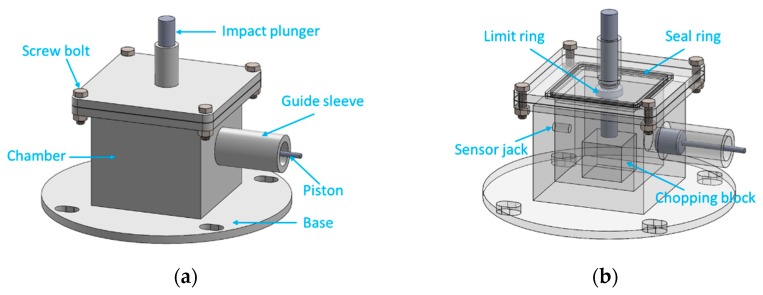
The engineering entity diagram and engineering perspective diagram of the energy release testing device: (**a**) engineering entity diagram; (**b**) engineering perspective diagram.

**Figure 11 polymers-11-00149-f011:**
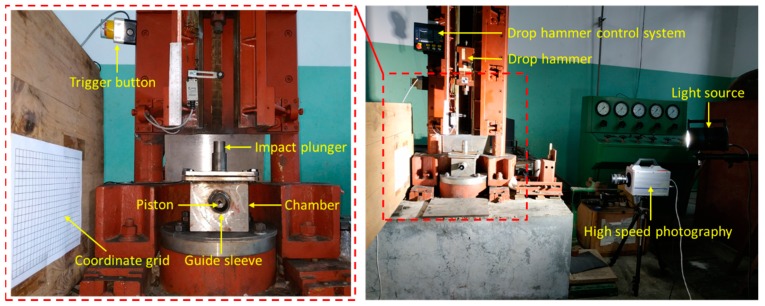
The layout diagram of the energy release testing device.

**Figure 12 polymers-11-00149-f012:**
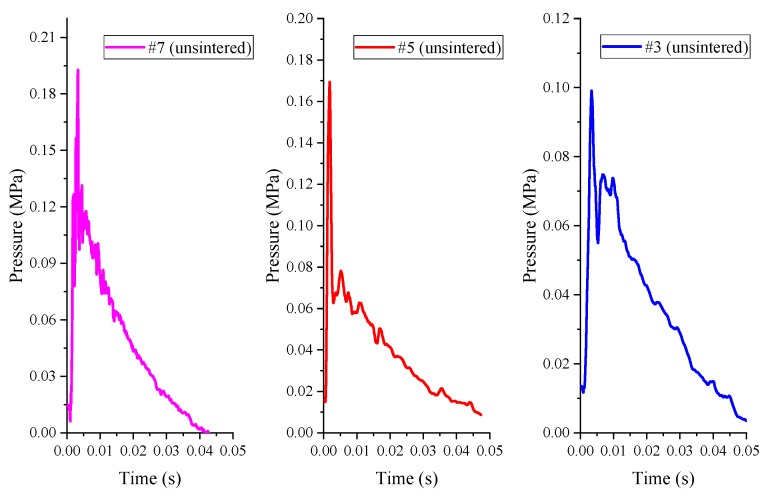
The pressure time history curve of the unsintered PTFE-based reactive materials.

**Figure 13 polymers-11-00149-f013:**
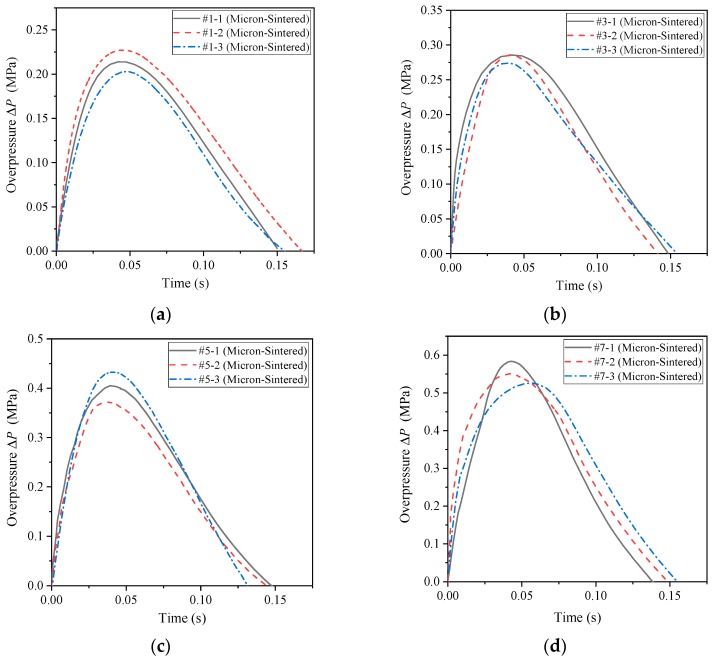
The pressure time history curve of the sintered micron-scale PTFE-based reactive materials: (**a**) #1 sintered reactive materials; (**b**) #3 sintered reactive materials; (**c**) #5 sintered reactive materials; (**d**) #7 sintered reactive materials.

**Figure 14 polymers-11-00149-f014:**
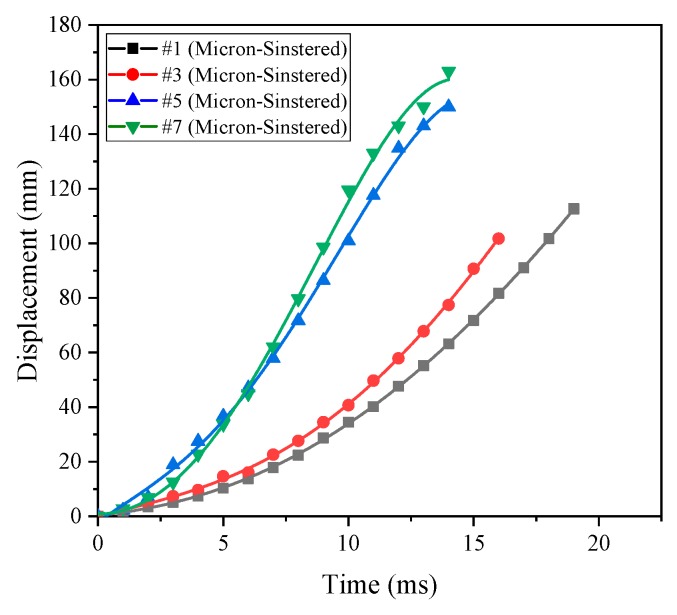
The piston displacement time curve corresponding to different formulations (#1, #3, #5, #7).

**Figure 15 polymers-11-00149-f015:**
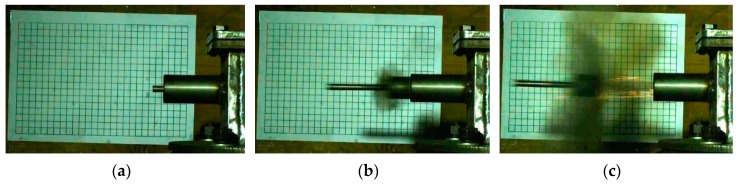
The piston motion state at different moments corresponding to sintered formulation #7: (**a**) initial state; (**b**) the state when the piston just flew out of the guide sleeve; (**c**) the piston flying through the air.

**Figure 16 polymers-11-00149-f016:**
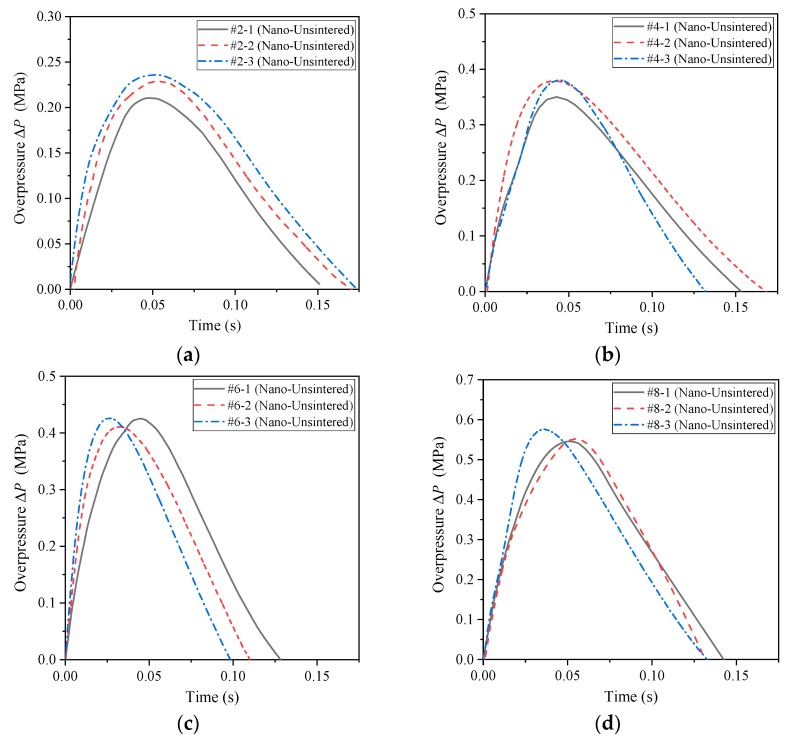
The pressure time history curve of the unsintered nano-scale PTFE-based reactive materials: (**a**) sintered reactive materials #2; (**b**) sintered reactive materials #4; (**c**) sintered reactive materials #6; (**d**) sintered reactive materials #8.

**Figure 17 polymers-11-00149-f017:**
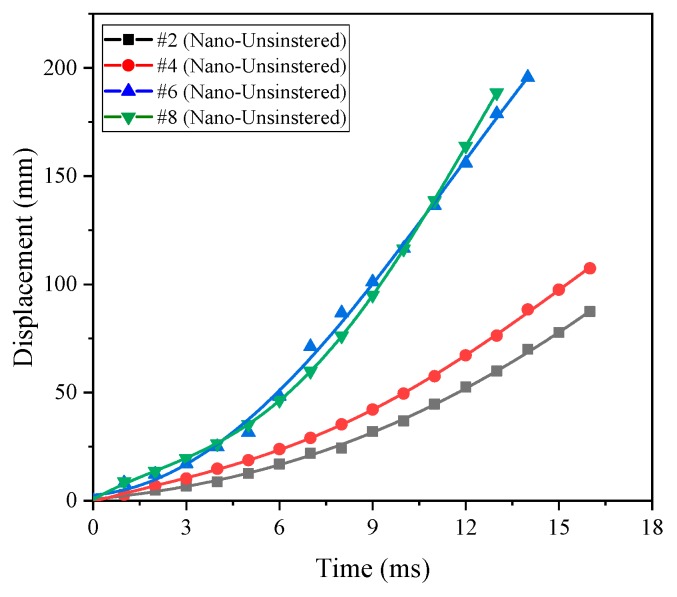
The piston displacement time curve corresponding to different formulations (#2, #4, #6, #8).

**Figure 18 polymers-11-00149-f018:**
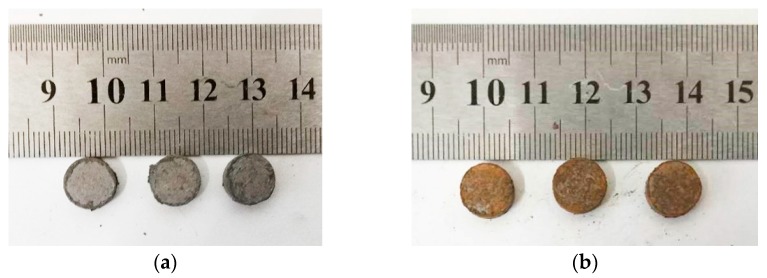
The sintered nano-scale PTFE-based reactive material sample: (**a**) formulation #2; (**b**) formulation #8.

**Figure 19 polymers-11-00149-f019:**
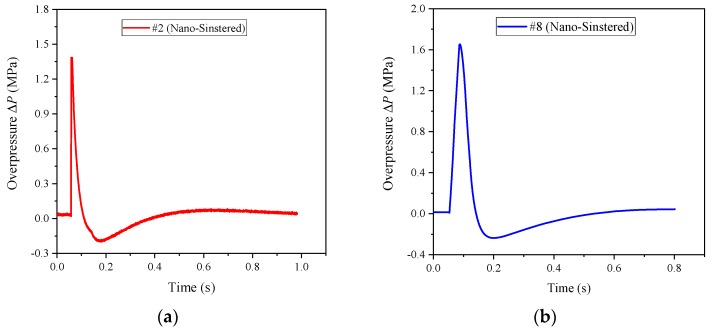
The pressure time history curve of the sintered nano-scale PTFE-based reactive materials: (**a**) sintered reactive materials #2; (**b**) sintered reactive materials #8.

**Figure 20 polymers-11-00149-f020:**
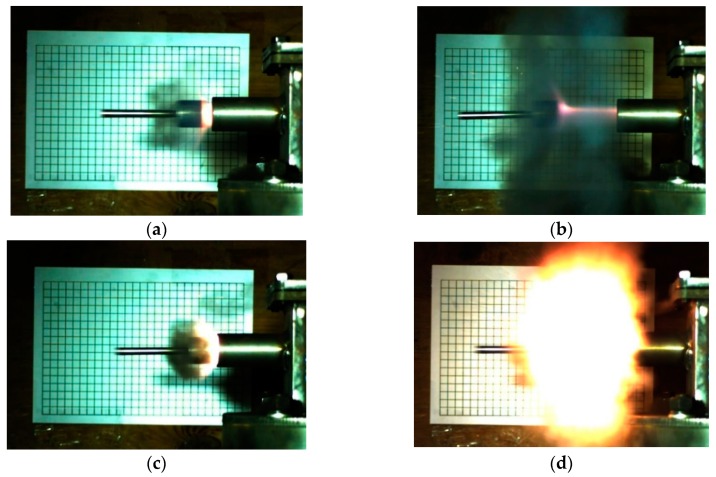
The piston motion state at different moments corresponding to sintered formulations #2 and #8: (**a**) the piston just flew out of the guide sleeve (#2); (**b**) the piston flies to the middle of the grid target (#2); (**c**) the piston just flew out of the guide sleeve (#8); (**d**) the piston flies to the middle of the grid target (#8).

**Table 1 polymers-11-00149-t001:** The formulations of reactive materials.

Formulation Number	PTFE/Al (μm) (wt %)	PTFE/Al (nm) (wt %)	Al/CuO (nm) (wt %)
#1	100		
#2		100	
#3	87.5		12.5
#4		87.5	12.5
#5	75		25
#6		75	25
#7	50		50
#8		50	50

Note: PTFE:Al = 73.5:26.5 CuO:Al = 81.5:18.5.

**Table 2 polymers-11-00149-t002:** The energy release rates of the four sets of sintered formulations (#1, #3, #5, #7).

Formulation Number	$\bar{\Delta P}$ (MPa)	$\Delta P^{*}$ (MPa)	*η* (%)
#1	0.215	3.494	6.15
#3	0.282	3.266	8.63
#5	0.403	3.038	13.27
#7	0.534	2.582	20.68

**Table 3 polymers-11-00149-t003:** The energy release rate of the four sets of unsintered formulations (#2, #4, #6, #8).

Formulation Number	$\bar{\Delta P}$ (MPa)	$\Delta P^{*}$ (MPa)	*η* (%)
#2	0.225	3.494	6.44
#4	0.370	3.266	11.33
#6	0.422	3.038	13.89
#8	0.558	2.582	21.61

**Table 4 polymers-11-00149-t004:** The energy release rates of the four sets of sintered formulations (#2, #4, #6, #8).

Formulation Number	$\bar{\Delta P}$ (MPa)	$\Delta P^{*}$ (MPa)	*η* (%)
#2	1.384	3.494	39.61
#4	1.446	3.266	44.27
#6	1.569	3.038	47.49
#8	1.643	2.582	63.63

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
