# Peer review of "Impact Energy Release Characteristics of PTFE/Al/CuO Reactive Materials Measured by a New Energy Release Testing Device"

_polymers, 2019, doi:10.3390/polym11010149_

Round 1

Reviewer 1 Report

The paper presents interesting experimental results related to  preparation and testing of reactive materials.  It can be recommended for publication after some polishing of the English.

Some examples:

1. The sentence in the Abstract: “The energy release rate of the nano-scale reactive materials is higher than that of the micron-scale reactive materials, because the reduction of particle size makes the particles have larger specific surface area, and the initial reaction energy required is lower” might be rewritten as following:

“The energy release rate of the nano-scale reactive materials is higher than that of the micron-scale reactive materials, because the reduction of particle size results in larger specific surface area.  Thus the energy required for ignition is lower”.

2. The next sentence “The energy release rate of sintered reactive materials is higher than that of unsintered reactive materials, because the interface between Al particles and PTFE particles in sintered reactive materials is larger, which makes the reaction more sufficient” might be rewritten as following:

“The energy release rate of sintered reactive materials is higher than that of unsintered reactive materials, because the interfacial area between Al particles and PTFE particles in sintered reactive materials is larger, which makes the reaction more sufficient”.

3. The words “initially recognized” in the second line of the Introduction should be replaced with the word “introduced”.

4. The sentence in the Introduction, page 1 “There are many types of reactive materials, among which the typical impact reactive materials are mainly three types of intermetallic compounds, thermite and metal/polymer mixtures” might be rewritten as following: “There are many types of impact reactive materials, the typical examples are intermetallic compounds, thermite and metal/polymer mixtures”. 

5. The next sentence “They differ greatly in reaction form, energy release effect and preparation methods” might be rewritten as following: “They differ greatly in reaction mechanisms, energy release and preparation methods”.

6. The sentence “Active metal particle-reinforced fluoropolymer materials, as a typical mixture of metal/polymer materials, have received special attention in recent years, and their superiority has also been verified in the application of weapons” might be rewritten as following: “A typical mixture of metal/polymer materials represented by active metal particle-reinforced fluoropolymer materials have received special attention in recent years resulting in their weapon applications”.  This statement should be accompanied by reference to publications in open literature, if available.

7. Sentence in the 2nd page “Joshi [7] proposed a preparation process of PTFE/Al material in 2003 in the form of patent, which enabled the prepared material to withstand a sufficiently large launch overload without breaking” might be rewritten as following “Joshi patented in 2003 a preparation process of PTFE/Al material with increased strength allowing to withstand a sufficiently large launch overload without breaking”.

8. The sentence in the page 3 “In 2007, Ames [28] developed his test method as a standard test method and published it to the public” might be rewritten as following “In 2007, Ames [28] introduced and published in open literature a standard test method”.

Author Response

Dear reviewer,

I am very grateful to you for reviewing my manuscript, and thank you for your affirmation and recognition of the manuscript. According to the suggestions you put forward and the opinions of the other reviewer, many changes have been made to the manuscript, including grammar, technical terms, and the place where additional references are needed is also supplemented.

As a non-English-speaking country scholar, the quality of written English does need to be improved. Therefore, I made a serious revision and applied for the MDPI English language editing (English editing ID: English-7429). The text has been checked for correct use of grammar and common technical terms, and edited to a level suitable for reporting research in a scholarly journal. I hope that the finishing manuscript can meet your requirements in terms of writing, and I will enhance my English writing ability in the future writing process.

Thanks very much for your attention to our manuscript.

Sincerely yours

Liangliang Ding

Reviewer 2 Report

Liangliang Ding et al presentthe Impact of  Energy Release  of 3 PTFE/Al/CuO by A 4 New Energy Release Testing Device. this work is interesting and the results are originals. HOWEVER, the manuscript structure is very poor and required reorganization, especially the english, just an example in the abstract: very long phrase without sens"The results show that with the increase of Al/CuO thermite content, the energy release rate of the reactive material increases obviously,  because the reaction threshold of Al/CuO is low and the heat generated can promote the reaction of  PTFE/Al."?!

In five ligne there are three "because"! repetition and please avoid to use the type of the non scientific word and changes it by : due, explained by, attributed to, etc....

I will be pleased to reconsider my décision if authors improves significantly the language level.

Author Response

(The authors gave the same response as above.)

Round 2

Reviewer 2 Report

Now, the revised paper can be accepted as is.